# A One Health Review of Community-Acquired Antimicrobial-Resistant *Escherichia coli* in India

**DOI:** 10.3390/ijerph182212089

**Published:** 2021-11-18

**Authors:** Keerthana Rajagopal, Sujith J. Chandy, Jay P. Graham

**Affiliations:** 1Berkeley School of Public Health, University of California, 2121 Berkeley Way, Room 5302, Berkeley, CA 94720, USA; kee_raja@berkeley.edu; 2Department of Pharmacology and Clinical Pharmacology, Christian Medical College, Vellore 632002, India; sjchandycmc@gmail.com

**Keywords:** community-acquired, antimicrobial resistance, *Escherichia coli*, One Health, India

## Abstract

Antimicrobial resistance (AMR) threatens to undermine nearly a century of progress since the first use of antimicrobial compounds. There is an increasing recognition of the links between antimicrobial use and AMR in humans, animals, and the environment (i.e., One Health) and the spread of AMR between these domains and around the globe. This systematic review applies a One Health approach—including humans, animals, and the environment—to characterize AMR in *Escherichia coli* in India. *E. coli* is an ideal species because it is readily shared between humans and animals, its transmission can be tracked more easily than anaerobes, it can survive and grow outside of the host environment, and it can mobilize AMR genes more easily than other intestinal bacteria. This review synthesized evidence from 38 studies examining antimicrobial-resistant *E. coli* (AR-E) across India. Studies of AR-E came from 18 states, isolated from different sample sources: Humans (*n* = 7), animals (*n* = 7), the environment (*n* = 20), and combinations of these categories, defined as interdisciplinary (*n* = 4). Several studies measured the prevalence of AMR in relation to last-line antimicrobials, including carbapenems (*n* = 11), third-generation cephalosporins (*n* = 18), and colistin (*n* = 4). Most studies included only one dimension of the One Health framework, highlighting the need for more studies that aim to characterize the relationship of AMR across different reservoirs of *E. coli*.

## 1. Introduction

Antimicrobial resistance (AMR) has been recognized as a major global health threat. Recent predictions estimate that deaths caused by AMR could reach 10 million by 2050 [1]. Recognizing the scale and importance of this global issue, the World Health Organization (WHO) has initiated a Global Action Plan to mitigate the effects of AMR. A key component of the Global Action Plan is to improve AMR surveillance capacity, especially in low- and middle-income countries (LMICs), with a One Health approach [2]. One Health is an approach that recognizes the interconnections between people, animals, and the environment [3]. It is applicable to understanding *Escherichia*
*coli (E. coli)* because of its prevalence and high transmission rates between humans, animals, and environmental interfaces. We focus on the role of AMR in *E. coli* because it is perhaps the most studied indicator organism, and its transmission can be tracked more easily (among animal hosts) than anaerobes [4,5,6,7]. Furthermore, *E. coli* can survive and even grow in the environment outside of the host, and may mobilize AMR genes more easily than other intestinal bacteria (such as Bacteroides) [8,9,10,11,12,13,14]. Many strains of *E. coli* have acquired the virulence genes necessary to cause a wide spectrum of intestinal and extra-intestinal infections such as diarrhea, urinary tract infections, and both community- and hospital-acquired bacteremia.

Poverty, population density, and high levels of infectious diseases combined with weakly implemented regulation and enforcement of antimicrobial use has led to a rapid increase in AMR in LMICs. In India, antimicrobial-resistant neonatal infections are estimated to result in 60,000 newborn fatalities each year [15]. It is likely that the source of many of these infections is rooted in the community, and thus will require a multifaceted One Health approach to detect, prevent, and control AMR. The presence of drug-resistant community-acquired infections is due to a diverse set of interrelated mechanisms, including inappropriate antimicrobial use, availability of antimicrobials on the open market, use in food animal production, and the current state of environmental pollution in both potable and surface waters [16,17,18]. Thus, AMR is an exemplary One Health challenge, as resistant microorganisms spread between people, animals, and the environment [3].

The Government of India has launched a five-year National Action Plan (2017–2021) to combat AMR. Under the National Action Plan, the Indian Council of Medical Research (ICMR) has established the National Anti-Microbial Resistance Research and Surveillance Network (AMRRSN) to compile national antimicrobial resistance data, which is key to identifying the root causes of the problem and creating effective interventions.

Most studies that have investigated the magnitude of AMR have focused on nosocomial infections and have been carried out predominantly in hospital settings [19,20,21]. Studies that examine AMR in aquatic environments, food, animals, and animal products have historically been narrow in their approach and have not attempted to link the AMR research in those domains to AMR in humans.

This systematic review identifies relevant research across India that characterizes AR-E from the environment, non-clinical human samples (i.e., community-acquired AR-E), and animal (domestic and wild) samples, where there is high human–animal or human–environment overlap with the potential for human exposure. *E. coli* is considered a good indicator for AMR surveillance [22,23]. Hence, the aim of this study was to analyze the available information on AR-E and determine the antimicrobial susceptibility patterns of *E. coli* isolated from different sources.

## 2. Materials and Methods

This review was conducted between May to September 2020 and followed the preferred reporting items for systematic reviews and meta-analysis (PRISMA) guidelines [24]. The study focused on the prevalence of AR-E in environmental samples of water obtained from lakes and rivers, of food animal sources (e.g., fish, chicken, vegetables, and bovine milk), and of community-acquired AMR in humans, and it targeted samples collected in community or outpatient settings (we excluded nosocomial AMR). We defined interdisciplinary studies as those that investigated the occurrence of AR-E in a combination of the two or three categories—environmental samples, animal food sources, and/or human samples.

PubMed was searched and the query terms used are outlined in the Appendix. No limit on publication dates was set. The database was queried on 19 June 2020. Results were imported into Covidence (www.covidence.org accessed on 20 June 2020), a systematic review management software, and duplicates were removed. All included studies focused on AR-E. Studies that included human isolates only assessed community-acquired infections in the human population. We searched PubMed (https://pubmed.ncbi.nlm.nih.gov, accessed on 19 June 2020) using the following terms pertaining to antimicrobial resistance: (“Enterobacteriaceae” OR “Gram negative bacteria” OR “*E. coli*” OR “*Escherischia coli*”) AND (“Drug Resistance” OR “Extended-spectrum beta-lactamase” OR “ESBL” OR “antibiotic” OR “antimicrobial” OR “AMR”) AND (“resistance” OR “resistant”) AND (“human” OR “community acquired” OR “community-acquired” OR “livestock” OR “poultry” OR “cattle” OR “cows” OR “pets” OR “chickens” OR “Environment” OR “water”) AND (“India”). We also searched the resulting reference lists to identify additional articles. The search terms are described in Table A1.

After the initial search, the research focus was narrowed further to include only *E. coli*. If studies examined other bacterial species along with *E. coli*, they were included. However, only information relevant to findings related to *E. coli* were included to improve comparability.

Of the 747 non-duplicate results, 613 studies were excluded based on the title and abstract screening process. Common reasons for exclusion included: (1) Focus was on human clinical AMR; (2) predominantly reported AMR in other species within Enterobacteriaceae and other Gram-negative bacteria; (3) study conducted outside of India; (4) study focused on isolation and molecular characterization of phages as opposed to AMR. Of the 134 selected articles, 96 articles were excluded after reviewing the full manuscript (see Figure 1). After a full-text review, 38 full texts were included for data extraction. Publications that described human or animal populations or environmental samples, bacteria isolates, and specific laboratory methods such as disk diffusion, antimicrobial susceptibility patterns, interpretation of resistance profiles of *E. coli*, and multidrug resistance (MDR) were considered and included in the study.

## 3. Results

The systematic review included studies from 18 states spread across the entire geography of the country. Thirty-eight studies were included in total, with samples collected from humans (*n* = 7), the environment (*n* = 20), animals (*n* = 7), and interdisciplinary sources (*n* = 4).

All AMR research studies (*n* = 38) applied culture-based antimicrobial susceptibility testing (AST) using the disk diffusion method. Studies (*n* = 29) also applied polymerase chain reaction (PCR) for AMR gene identification and assessing AMR mobile genetic elements. Studies that employed PCR provided estimates of resistance genes for the isolates identified.

In Figure 2, we tabulated the prevalence of antimicrobial resistance in *E. coli* for six drugs that have high clinical relevance: Ampicillin, amikacin, cephalosporins, ceftazidime, imipenem, and polypeptides that span five different classes of antimicrobials. In Figure 3, we map the location of the studies included in this review. 

The AR-E burden has serious implications for human health, owing to the potential transmission of bacteria from animals and the environment to humans, thereby impairing the efficacy of antimicrobial treatment and compromising public health. All included studies were in the English language and the key study findings are described in Table A2.

Below, we summarize the study results by the source from which the AR-E were derived.

### 3.1. AR-E in Humans

Seven studies examined AR-E in human isolates, three of which used human stool samples and the other four used urine samples. These studies reported the prevalence of community-acquired AMR (i.e., no hospital-acquired infections were included). Of these, three studies reported the highest susceptibility of *E. coli* to amikacin [25,26,27]. In New Delhi, one study examined the prevalence of AMR to β-lactams in neonates (less than 60 days old) with no history of antimicrobial use. Nearly 87% of *E. coli* from neonates were resistant to ampicillin [28]. This study was the first to demonstrate the load of community-acquired beta-lactamase-producing *E. coli* and suggested that the babies likely acquired these strains from the maternal flora.

### 3.2. AR-E in Animals

Seven studies included in this research examined the presence of *E. coli* in samples derived from animals that are mainly consumed as food and constitute an important part of the human diet. The detection of ESBL-producing *E. coli* was conducted in three of these studies, and the results confirmed ESBL production as follows: 54.5% *E. coli* isolated from milk (*n* = 22 isolates) [29]; 25.4% *E. coli* isolated from pig feces (*n* = 867 isolates) [30]; 71.6% *E. coli* isolated from fresh seafood samples (*n* = 475 isolates) [31]. In a study conducted by Vinayananda et al., the researchers sampled 840 eggs available in markets across southern India. The samples included table eggs from three sources: Processed commercial layer farms, unprocessed commercial layer farms, and free-range eggs collected from household backyards. *E. coli* was present in 28.5% of the overall samples with an occurrence rate of 22.9, 29.2 and 50.0% from processed, unprocessed and free-range table eggs, respectively. The study recovered 24 isolates, in which 100% were resistant to cloxacillin, irrespective of the source from where the eggs where obtained. The multiple antimicrobial resistance (MAR) index was calculated and interpreted according to Krumperman (1983) using the formula: a/b, where “a” represents the number of antimicrobials to which a particular isolate was resistant and “b” the total number of antimicrobials tested. The pattern of antimicrobial resistance was similar among the three sources, with an MAR index of 0.19 for free-range table eggs, 0.23 for unprocessed eggs, and 0.28 for processed eggs. The authors stated that the high MAR index demonstrated the need to control the use of antimicrobials in animal farming to prevent cross-transfer of ESBL-producing *E. coli* to humans, as eggs function as a staple in the Indian diet [32].

### 3.3. AR-E in the Environment

The majority of studies included in this systematic review focused on the presence of AR-E in the environment (*n* = 20). Sources were predominantly water samples from lakes, reservoirs, and river water, while some other studies included sources such as vegetables and fruits. Several studies focused on pathogenic forms of *E. coli* and examined the presence of Shiga toxin genes stx1 and stx2 [33,34,35,36,37,38]. The presence of virulence genes specific to certain *E. coli* pathotypes may cause severe health outcomes such as diarrhea, urinary tract infections, hemolytic colitis, neonatal meningitis, nosocomial septicemia, hemolytic uremic syndrome, and surgical site infections [39]. In a study that focused on determining the prevalence of resistant coliforms in the Yamuna river, 123 (86.6%) isolates showed resistance to three or more drug classes and were considered MDR. High resistance rates were observed for cefazolin (88.7%), followed by vancomycin (74.6%), cefuroxime and cefotaxime (60%), and gentamicin (52.8%). One fourth of the total isolates (*n* = 141) were resistant to 10 or more drugs [40].

### 3.4. AR-E in Interdisciplinary Studies

Four studies included in this review were classified as interdisciplinary. Only two studies, by Sahoo et al. and Puii et al., analyzed *E. coli* from all three dimensions of the One Health approach (i.e., environment, animals, and humans). Sahoo et al. investigated AR-E isolated from child stool samples, cow dung, and drinking water. Ninety percent of the isolates were resistant to at least one antimicrobial [41]. *E. coli* isolates from non-coastal regions exhibited significant resistance to second- and third-generation cephalosporins and nalidixic acid [41]. Rasheed et al. examined resistance in isolates of *E. coli* from vegetables and animal food sources (i.e., meat, eggs, and milk). The overall prevalence of drug-resistant *E. coli* was 14.7%. Pathogenic *E. coli* cycling through food is of particular concern, as it poses a significant health risk to humans [11].

## 4. Discussion

Among the studies reviewed, extensive resistance to ampicillin and third-generation cephalosporins (3GC) were observed, while carbapenem resistance was less common. Many studies discussed the importance of unrestricted access to over-the-counter antimicrobials as a factor amplifying the problem of AMR [28]. Fundamental change is required in the way that antimicrobials are prescribed, distributed, and consumed. Risk assessments conducted by the WHO show that the Southeast Asia region is likely to be the most at-risk part of the world [2]. In light of the recent global COVID-19 pandemic, experiences and consequences need to be translated into policy and on the ground action to prevent the more silent but rising AMR pandemic. The unprecedented speed at which new strains and “superbugs” develop and are transmitted globally needs to be met with an equally urgent response by the global community. AMR needs to be addressed as a serious and potentially catastrophic issue with global significance.

The presence of ESBL genes isolated from various sample sources is a great concern for human beings [28,29,42,43,44]. Drug resistance can be transferred between bacteria in the gut of humans and animals and this can greatly complicate treatment. The antimicrobial susceptibility profiles of *E. coli* isolates included in this review indicate that most of them have acquired multidrug resistance against a broad set of antimicrobial classes. In the studies we reviewed, the presence/absence of a resistance gene does not necessarily mean that the genes are expressed.

The emergence of multidrug-resistant *E. coli* isolates, involving co-resistance to four or more unrelated families of antimicrobials, is a serious concern, and many of the studies in this review demonstrated a high prevalence of MDR *E. coli* [45]. Verma et al. highlighted that *E. coli* strains from vegetables and fruits exhibited resistance to almost all classes of drugs, including quinolones, carbapenems, penicillin, and aminoglycosides. Furthermore, the review highlighted that many researchers apply different definitions in their AMR studies. For example, the definition of MDR varied across the studies. In some studies, MDR *E. coli* was defined as isolates resistant to five or more antimicrobials. For example, Sukumaran et al. reported 53.3% of *E. coli* serotypes showing resistance to five or more antimicrobials and classified them as MDR [46]. However, studies conducted by Abhirosh et al. and Mohanta and Goel defined MDR as those resistant to three or more antimicrobials [47,48]. This review demonstrated that more needs to be done in order to create standard definitions and protocols for studying AMR in India.

There is a lack of research on exposure assessment and exposure patterns to AR-E. There is a need for more quantitative microbial risk assessment in order to assess and objectively quantify the risk of human exposure to AR-E. There is a significant need to identify the highest risk sources (e.g., food or drinking water), geographic locations, and human–animal or human–environment interfaces that result in the greatest risk of humans to be exposed to AMR.

The prevalence of antimicrobial-resistant infections caused by extended spectrum β-lactamase (ESBL)-producing *E. coli* has increased over the last decade across every geographic region of the world.This study highlights the presence of diverse reservoirs of drug-resistant *E. coli.* Most studies are not able to confirm whether the increase in the number of resistant strains isolated in one medium such as water necessarily causes an increase in the number of strains isolated from animals consuming this water.

Only ~10% (*n* = 4) of the studies included in this review analyzed *E. coli* from a combination of the three dimensions of the One Health approach (i.e., environment, animals, and humans). Refer Table A2 for specific details and a summary of each study [8,11,16,25,26,27,28,29,30,31,32,33,34,35,36,37,38,40,41,42,43,44,46,47,48,49,50,51,52,53,54,55,56,57,58,59,60,61]. One Health is a powerful transdisciplinary tool. However, the existing fragmentation and lack of true partnerships among the separate entities and actors that regulate animal health, human health, and environmental protection leads to difficulties in the actual implementation of a One Health approach. Appropriate antimicrobial stewardship is paramount to preserving the value of this resource and will require an integrated effort across multiple disciplines in India.

## 5. Conclusions

The prevalence of resistance to last-resort antimicrobials such as colistin and imipenem increases treatment complications. This should be viewed seriously from a population outcomes perspective, as these classes of drugs are categorized as lifesaving and used in treating serious infections. In India, countrywide measures for the surveillance of AMR needs to be strengthened and expanded. Surveillance should incorporate both environmental samples and clinical isolates from humans and animals in order to better determine the sources and drivers of AMR.

The prevalence of AR-E across various interfaces in the community is a reminder of the intimate and delicate relationship between humans, animals, and the environment. The studies included in this review individually highlight the prevalence of AMR across different areas of India and across multiple sources. There was difficulty, however, in making comparisons between the results of included studies given the diverse settings, methods used, and sources of samples. More research should apply a multidisciplinary methodology focusing on the interfaces between AMR in humans, the environment, food and animals, and/or social–ecological systems. We also need more microbial risk assessment research that aims to define the root causes of AMR and the factors that increase the risk of human exposure to AMR. Finally, there is a need to develop feasible and sustainable public health interventions that encompass the One Health approach and focus on developing a holistic solution to AMR.

## Figures and Tables

**Figure 1 ijerph-18-12089-f001:**
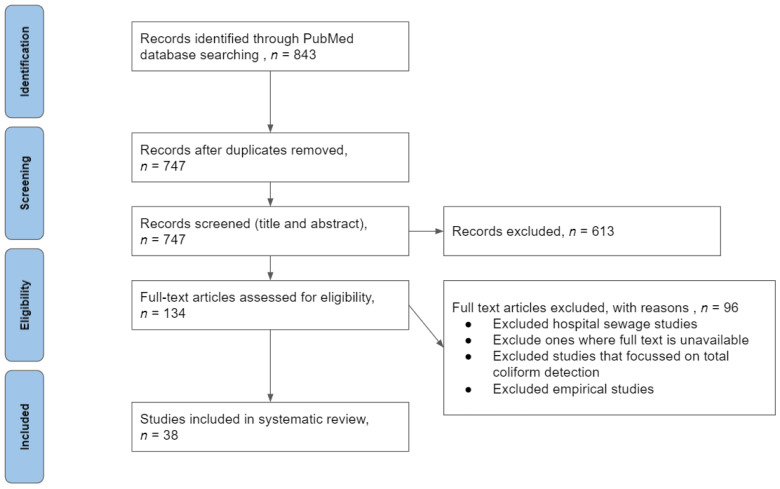
PRISMA flowchart of the systematic review process and article screening results.

**Figure 2 ijerph-18-12089-f002:**
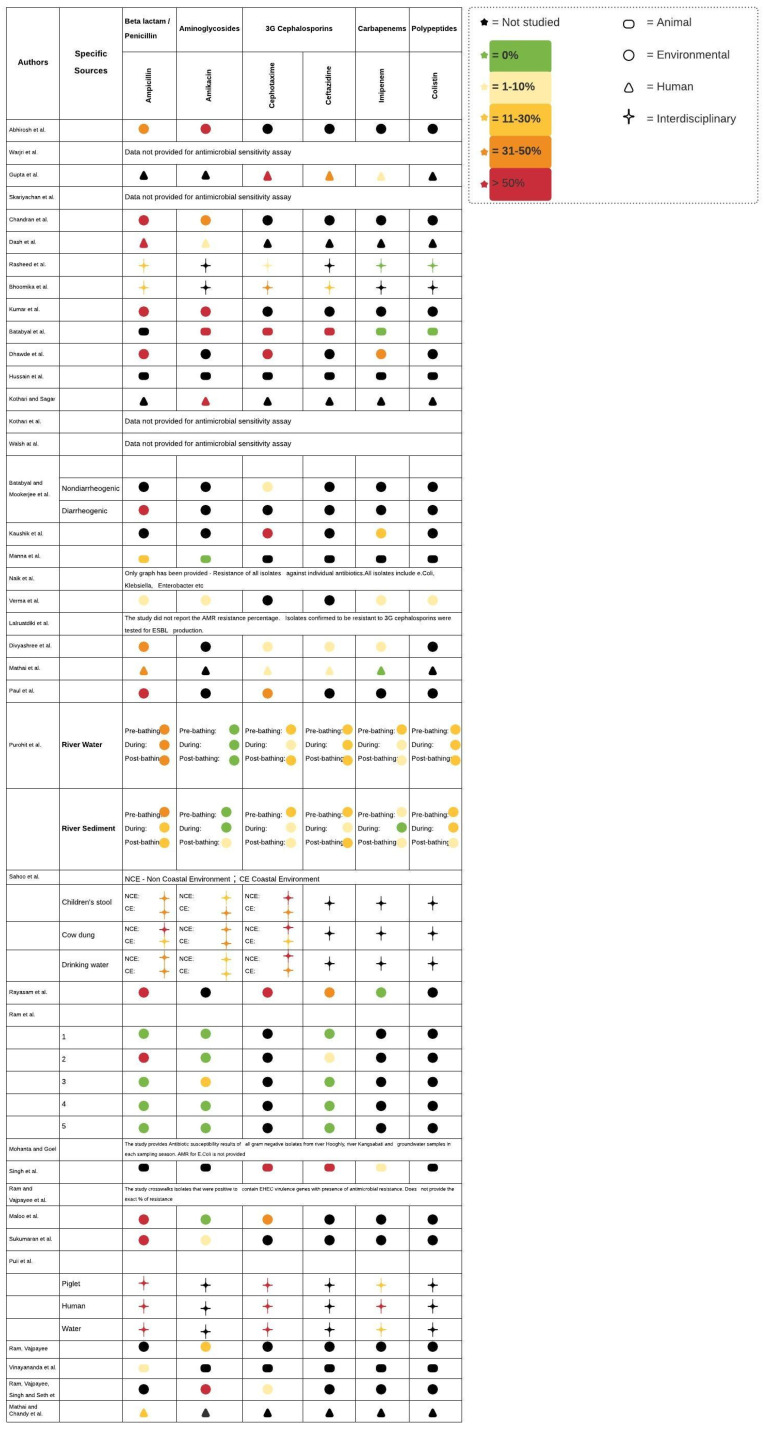
Antibiotic susceptibilities of *E. coli* in India—a review of the literature. This figure tabulates the prevalence of antimicrobial resistance in *E. coli* for six drugs that have high clinical relevance. The color codes indicate the percent resistance of isolates against individual antibiotics. The shape represents the source from which the AR-E were derived.

**Figure 3 ijerph-18-12089-f003:**
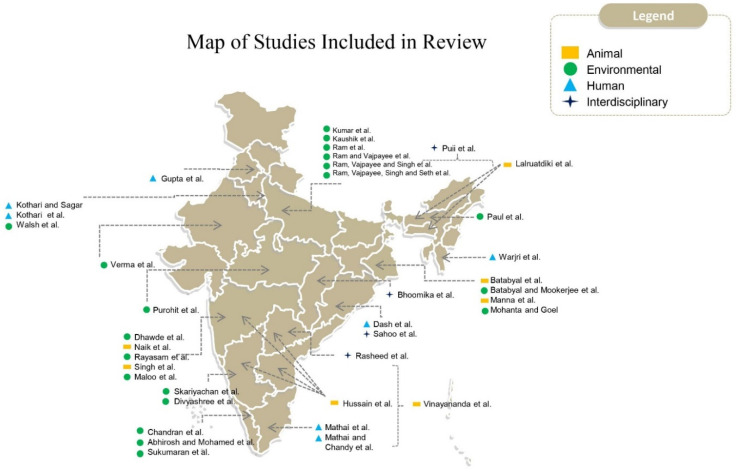
Map of the location of the studies and the focus of the study included in this review.

## Data Availability

All articles and reports used for this review are available to the public.

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
