# Peer review of "A One Health Review of Community-Acquired Antimicrobial-Resistant Escherichia coli in India"

_ijerph, 2021, doi:10.3390/ijerph182212089_

Round 1
Reviewer 1 Report
The work presented to me for the review addresses a global problem,the spread of multidrug resistant strains (E. coli) in the environment.
For this purpose, the PubMed database was searched.
However, I have doubts as to whether this search was carried out correctly.
I am concerned that the renaming from Enterobacteriaceae to Enterobacterales
has not been taken into account (only Enterobacteriaceae results are included).
Moreover, the strains were found on the basis of the PCR test. Which is not
right. Because this only tells us about the presence of the gene, not its
activity. How did the authors confirm the information that the genes
responsible for specific resistances were active? Their detection is not
indicative of resistance. Please specify this information in the manuscript.
Omitting this important information may affect further analysis.
It is also unclear to me why the authors of the names of antibiotics sometimes
write with a capital letter and sometimes with a lower case. Please clarify.
Author Response
Dear colleagues, I have attached our responses to the review comments

Reviewer 2 Report
Dear Authors,
it is a very interesting research work. Thank you for the opportunity to read it.
General thoughts:
Figure 2 - I would consider dividing it into two pages due to its low readability.
Discussion
I understand that the work is only about India.
Has there been an increase in the prevalence of resistant strains over the years? The works used in the publication as the "core" are from different years. Such a reference would be a good argument for the One Health concept. Is there any correlation between the increase in the number of resistant strains isolated, for example, in water, and the number of strains isolated, for example, in meat?
And is it possible to compare it with the situation in, for example, Europe?
Author Response
We have attached our responses to the reviewer's comments.
